# Variance-Aware Machine Translation Test Sets

**Runzhe Zhan**[1*] **Xuebo Liu**[2*] **Derek F. Wong**[1†] **Lidia S. Chao**[1]

[1]NLP[2]CT Lab, Department of Computer and Information Science, University of Macau
nlp2ct.runzhe@gmail.com, {derekfw,lidiasc}@um.edu.mo
[2]Institute of Computing and Intelligence, Harbin Institute of Technology (Shenzhen), China
liuxuebo@hit.edu.cn

## Abstract

We release 70 small and discriminative test sets for machine translation (MT) evaluation called *variance-aware test sets* (VAT), covering 35 translation directions from WMT16 to WMT20 competitions. VAT is automatically created by a novel *variance-aware filtering method* that filters the indiscriminative test instances of the current MT test sets without any human labor. Experimental results show that VAT outperforms the original WMT test sets in terms of the correlation with human judgement across mainstream language pairs and test sets. Further analysis on the properties of VAT reveals the challenging linguistic features (e.g., translation of low-frequency words and proper nouns) for competitive MT systems, providing guidance for constructing future MT test sets. The test sets and the code for preparing variance-aware MT test sets are freely available at https://github.com/NLP2CT/Variance-Aware-MT-Test-Sets.

## 1 Introduction

Automated machine translation (MT) evaluation relies on metrics and test sets. Based on the use of test sets, the metrics to quantify the performance of MT systems can be divided into two categories: reference-based metrics (Papineni et al., 2002; Popović, 2015; Lo, 2019; Zhang et al., 2020) and reference-free metrics (Popović, 2012; Yankovskaya et al., 2019). Reference-based metrics which measure the overlap between the reference and model's hypothesis, are widely used both in research and practice. Even the state-of-the-art metrics that exploit a pre-trained model (Zhang et al., 2020; Sellam et al., 2020; Rei et al., 2020) are able to evaluate the finer-grained semantic overlap, it still cannot achieve human-level judgements (Ma et al., 2019; Mathur et al., 2020). Although the metric itself can be further elaborated, the reference in the test set, which is another key ingredient in the MT evaluation, has received less attention from the community.

The references are not innocent of confusing automatic metrics. Research has proven that the collected references tend to exhibit a monotonous translation style (Popovic, 2019; Freitag et al., 2020b) instead of natural text, and lack diversity in the evaluation. On the other hand, the competitive MT systems typically share a homogeneous architecture and training data, causing the performance of the MT systems to be too close to be distinguished, thus the differences in the scores given by the automatic metrics are small. To alleviate this problem, previous work has focused on increasing the diversity of the references by means of paraphrasing, including human paraphrasing (Freitag et al., 2020a,b) and automatic paraphrasing (Kauchak and Barzilay, 2006; Guo and Hu, 2019; Bawden et al., 2020), but both of these are expensive in terms of human labor and computational cost. Considering the fact that not all the references are monotonous, it is still unclear how to select those discriminative references instead of diversifying them for the MT evaluation.

---

[*] Equal contribution

[†] Corresponding author

35th Conference on Neural Information Processing Systems (NeurIPS 2021) Track on Datasets and Benchmarks.

This paper aims to tackle this problem without any human labor. Our motivation comes from a common fact in the real world. In a general examination or test, the simplest and most difficult questions cannot tell the difference between the examinees because they may be all correctly or incorrectly answered. Accordingly, those questions that receive diverse answers play a vital role in distinguishing the examinees' abilities by comparing them with the ground truth. Based on this fact, a similar phenomenon can also happen in the test set for evaluating machine translations.

In this paper, we use the variance of translation scores evaluated by the metric as a criterion to create a *variance-aware* test set whose references are more discriminative in evaluating MT systems. The selected references are characterized by their diverse evaluation scores, indicating that the MT systems are not consistent in translating the same source, thus this translation case is a valuable indicator for distinguishing the capability of MT systems. Experimental results show that evaluating with the created *variance-aware* test set can improve the correlation with human judgements. Further analysis of the properties of the *variance-aware* test set also confirms its effectiveness.

Our main contributions are as follows:

- We release 70 *variance-aware* MT test sets, covering 35 translation directions from the WMT16 to WMT20 competitions. The test set filters 60% of the test instances from the original WMT version, which is time-efficient for research consuming high computational resources (e.g., reinforcement learning and neural architecture search).

- We propose a simple and effective method to automatically identify discriminative test instances from MT test sets. We demonstrate that using the discriminative test instances can yield a better correlation with human judgements than using the original test set.

- We give an in-depth analysis of the properties of discriminative and non-discriminative test instances. We find that the translations of low-frequency words and proper nouns are highly discriminative, providing clues for building challenging MT test sets.

## 2 Background

### 2.1 MT Evaluation and Meta-Evaluation

The evaluation of machine translation is a crucial topic in the development of MT due to the need to compare the performance of several candidate MT systems. Traditionally, human assessment is used to evaluate MT systems, but it is expensive in terms of its costs. Moreover, the quality of the assessment of crowdsourced evaluation work is unpredictable, and there is a big gap between non-expert and professional translators (Toral et al., 2018; Läubli et al., 2020; Mathur et al., 2020). Therefore, automatic evaluation metrics have received a lot of attention due to their advantages, such as their low cost and the controllability of the process, and are now widely used in model selection and optimization (Shen et al., 2016; Wieting et al., 2019).

The reference-based metrics which rely on a reference translation are the most popular automatic evaluation metrics. They differ in the ways they measure overlap. For example, BLEU (Papineni et al., 2002) and its variants (Doddington, 2002; Popović, 2015) evaluate the overlap by matching the $n$-grams, and other metrics like TER (Snover et al., 2006) quantify the overlap by the edit distance. However, these metrics are conducted in a hard matching paradigm and do not consider semantics. METEOR (Banerjee and Lavie, 2005; Denkowski and Lavie, 2014) alleviates this problem by introducing synonymy and other linguistic features in the word matching but is limited in the availability of language resources. Recent embedding-based metrics break the limitation of hard matching, making it possible to evaluate the semantic overlap. By enhancing the semantic representation by a pre-trained model (Devlin et al., 2019; Lample and Conneau, 2019), BERTScore (Zhang et al., 2020) correlates better with human judgements than previous metrics. At the same time, the end-to-end paradigm using a pre-trained representation is also applied to the evaluation of MT, and has achieved remarkable performance, e.g., COMET (Rei et al., 2020) and BLEURT (Sellam et al., 2020).

To verify the effectiveness of automatic evaluations, the process that measures the correlation between the scores given by an automatic metric and human ratings is called meta-evaluation. A meta-evaluation mainly uses correlation coefficients such as Pearson's $r$ to determine the extent to which the automatic metric performs like a human evaluator (Callison-Burch et al., 2006, 2008). The

validation process of our method covers the mainstream metrics and uses the ordinary meta-evaluation methods to validate the improvement in the correlation.

## 2.2 Shortcomings of Current Test Sets

The less discriminative instances in a public benchmark are the bottleneck of automatic evaluation. The test sets released by the organizers of the WMT competition are the well-recognized benchmark for MT evaluation. However, some have argued that some of the references in these test sets may mislead reference-based metrics, making the evaluation results of automatic metrics different from human judgement. One major issue is that the existing references tend to be monotonous (Popovic, 2019; Freitag et al., 2020b). This translation style is easy to achieve by the MT systems and less discriminative for the evaluation. In addition, based on the phenomenon observed by Zhan et al. (2021) in the evaluation of the WMT19 English→German task, most tokens can be correctly translated by all the participation systems, especially for the competitive ones, indicating that the test sets are only partly valuable in distinguishing the MT systems.

There are ways to create a more diverse test set so as to improve discernment. Kauchak and Barzilay (2006) explored the automatic paraphrasing techniques for improving the accuracy of automatic metrics and validated their effectiveness on small-scale human assessment data. Bawden et al. (2020) further investigated the use of automatic paraphrasing in automatic evaluation, finding limited gains in correlation with human judgements on the WMT19 benchmark. Promisingly, human paraphrased references have proved that they can significantly improve the correlation metrics of BLEU on some language pairs (Freitag et al., 2020b). Overall, these methods to augment the references are restricted by their construction costs and consistently limited improvement.

Instead of diversifying the references, our work pays attention to selecting the discriminative part from the existing test set for better distinguishing between strong MT systems.

## 3 Variance-Aware Test Set

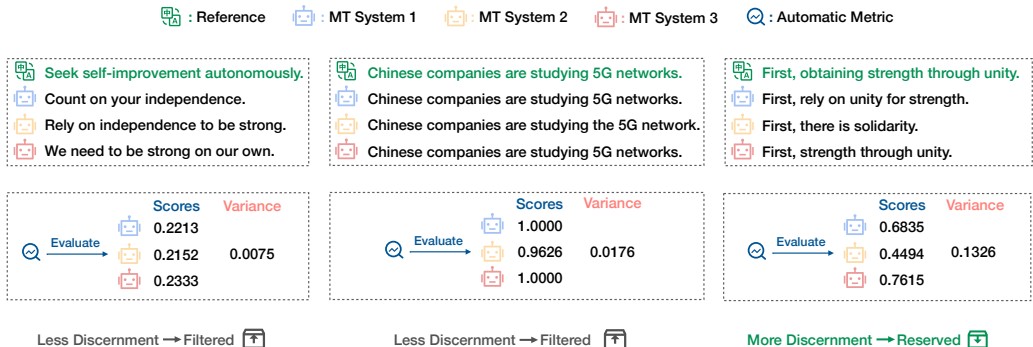

Figure 1: An illustration of the proposed variance-aware filtering method.

## 3.1 Motivation

Generally speaking, the test items that are either too easy or too difficult, cannot tell the differences of test-takers since they would not perform very differently when answering the extremely simple or difficult questions.

Machine translation evaluation is a kind of test. One can make an analogy between MT evaluation and tests in general: the MT systems are the test-takers, the instances in the test set are the test items. Similarly, to discriminate between MT systems in terms of their ability, a discriminative test instance must reveal clear differences in the systems' performance. As illustrated in Figure 1, evaluating MT systems' performance by using the first two references causes a subtle difference in evaluation results due to the polarized difficulty, thus it is hard to discriminate between the systems in this circumstance. By contrast, the gap of evaluation results in the last case is huge enough to detect the differences in translation ability.

These cases clearly show that a discriminative test instance must make the evaluation exhibit a large diversity so that it can become a decisive clue for comparing the MT systems. Since the metrics evaluate the performance of an MT system, the discrimination power of test instances can be quantified by the variance of scores given by a metric, reflecting the degree of diversity in the evaluation. A higher variance indicates that using this test instance in the evaluation makes it easier to differentiate between the systems. Therefore, our goal is to create a discriminative test set for better evaluating MT systems by selecting the instances whose variance of evaluation results is high. This process will be referred to as *variance-aware filtering*.

## 3.2 Variance-Aware Filtering

To measure how differently MT systems perform on a test instance, the performance of candidate systems is firstly quantified by the automatic metrics, then the standard deviation is simply used as a statistical indicator to model the diversity of the evaluated performance. The standard deviation takes the square root of the variance, we use it because the scale of this measurement is the same as the original data. Given $N$ references $\mathcal{T} = \{t_1, t_2, ..., t_N\}$ and a set of corresponding hypotheses $\mathbf{h} = \{\mathbf{h}_1, \mathbf{h}_2, ..., \mathbf{h}_N\}$ generated by $k$ systems in which $\mathbf{h}_i = \left\{h_i^{(1)}, h_i^{(2)}, ..., h_i^{(k)}\right\}$, the performance diversity of hypothesis $\mathbf{h}_i$ is estimated by the standard deviation $\sigma_i$ of the scores, which is formulated as:

$$\sigma_i = \sqrt{\frac{1}{k}\sum_{j=1}^{k}(\mathcal{M}(h_i^{(j)}, t_i) - \mu_i)^2}, \ 1 \leq i \leq N \tag{1}$$

where $\mathcal{M}(\cdot, \cdot)$ is the metric used to score the performance of the translation and $\mu_i$ is the average value of all the systems' scores, which can be calculated as follows:

$$\mu_i = \frac{1}{k}\sum_{j=1}^{k}\mathcal{M}(h_i^{(j)}, t_i), \ 1 \leq i \leq N \tag{2}$$

For all the standard deviations $\{\sigma_1, \sigma_2, ..., \sigma_N\}$, a higher $\sigma_i$ indicates that the behaviour of the systems on reference $t_i$ is more diverse. Therefore, $\lambda$ percent of test instances whose corresponding references have lower values of $\sigma$ will be filtered out in order to create a new discriminative test set, where $\lambda$ is a hyperparameter determined by the empirical experiments.

## 4 Experiments and In-Depth Analysis

### 4.1 Experimental Setup

Table 1: Detailed information about the test sets involved in the experiments, where Num denotes the number of translation directions.

|  | **WMT16** (Num=7) | **WMT17** (Num=14) | **WMT18** (Num=14) | **WMT19** (Num=18) | **WMT20** (Num=17) |
|---|---|---|---|---|---|
| **X-English** | cs, de, fi, ro, ru, tr | cs, de, fi, lv, ru, tr, zh | cs, de, et, fi, ru, tr, zh | de, fi, gu, kk, lt, ru, zh | cs, de, iu, ja, km, pl, ps, ru, ta, zh |
| **English-X** | ru | cs, de, fi, lv, ru, tr, zh | cs, de, et, fi, ru, tr, zh | cs, de, fi, gu, kk, lt, ru, zh | cs, de, ja, pl, ru, ta, zh |
| **Others** | / | / | / | de-cs, de-fr, fr-de | / |

**Data**  Five WMT test sets (Bojar et al., 2016, 2017; Ma et al., 2018, 2019; Mathur et al., 2020) ranging from WMT16 to WMT20 were used to conduct the experiments since they are the well-recognized benchmarks in the MT community. The included translation directions are as shown in Table 1. On the other hand, we choose the test sets starting from WMT16 because the neural machine translation (Bahdanau et al., 2015; Sennrich et al., 2016; Vaswani et al., 2017) paradigm has largely improved the capability of MT systems and the systems submitted to the WMT competitions have gradually become more competitive since 2016. For each language pair, we use the raw data released

by the WMT competitions including the official references, submitted hypotheses of the different MT systems, and the corresponding human ratings.

**Metrics and Meta-Evaluation**    Without loss of generality, we validate our research hypotheses on the following four representative metrics, and use their public open-source implementations so that the results can be easily reproduced:

- **BLEU** (Papineni et al., 2002) is an $n$-gram based metric that uses the precision rate to evaluate the coverage of reference $n$-gram in the model hypothesis. We use the sentence-level BLEU in the filtering procedure and evaluate the corpus-level system performance.

- **COMET** (Rei et al., 2020) is an end-to-end metric that builds on the top of the pre-trained XLM model including reference-based models and reference-free models. We use the recommended reference-based estimator model in the experiments.

- **BLEURT** (Sellam et al., 2020) is an end-to-end metric that fine-tunes the BERT model with several regression and classification tasks to make the model better be adapted to the MT evaluation scenario. We use its released checkpoint and default settings in the experiments.

- **BERTScore** (Zhang et al., 2020) is an embedding-based metric that relies on a pre-trained BERT model to encode the reference and hypothesis, measuring the similarity of representation with precision (BERTS-P), recall (BERTS-R), and the $F$-measure (BERTS-F). For the evaluation of different language pairs, we use the default BERT-family models as the same as the BERTScore implementation, e.g., RoBERTa-large for evaluating English text and BERT-base-multilingual-case for evaluating other languages.

To examine the effectiveness of the automatic evaluation metrics, we use the system-level Pearson's $r$, Kendall's $\tau$ and Spearman's $\rho$ correlation coefficients as the metrics for measuring how the results of the automated evaluation correlate with human judgements; these are also widely used in the competitions (Macháček and Bojar, 2013; Mathur et al., 2020) and related research (Freitag et al., 2020b).

## 4.2  Ablation Study

There are two main factors that may affect our proposed filtering approach: the filtering percentage $\lambda$ and the filtering metric $\mathcal{M}$. Hence, a series of empirical experiments was conducted on the WMT20 benchmark to explore the best settings for building the most discriminative test sets, and the finalized setting would subsequently be used to validate its generality in other WMT benchmarks.

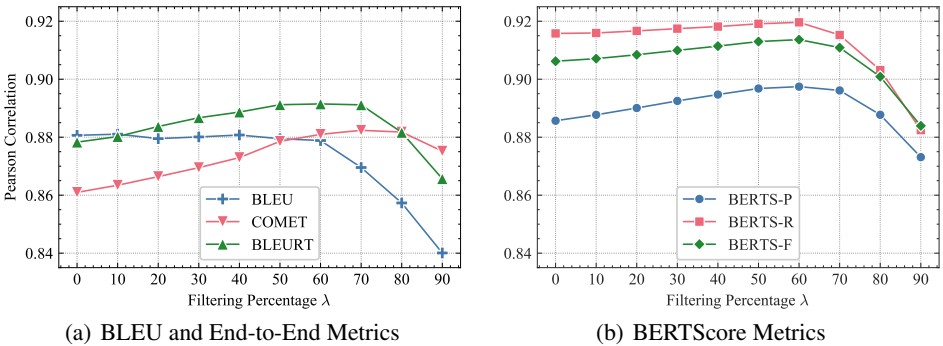

|                                   |                              |
| :-------------------------------: | :--------------------------: |
| (a) BLEU and End-to-End Metrics   | (b) BERTScore Metrics        |

Figure 2: Comparison of averaged Pearson correlations measured on all the WMT20 translation directions using different filtering percentages. Extreme settings will hurt the correlation results whereas filtering 60% or 70% of the data out of the test sets is the appropriate choice.

**Choice of Filtering Percentage** $\lambda$    affects the amount of data to be preserved and is also an indicator that reflects the discernment of the current data sets in terms of the evaluation metrics. As shown in Figure 2, using only a partial test set can improve the evaluation correlation of automatic metrics, but the most effective percentage setting depends on the type of evaluation metric. Compared to the

BLEU metric, the metrics driven by the pre-trained models achieve the local optimal correlation using a smaller proportion of the test set, i.e., $\lambda \geq 50$. The underlying reason for this may lie in the granularity of the evaluation in terms of the semantics: a metric that is better in parsing the semantics needs fewer data to distinguish the MT systems because of the larger impact of those discriminative samples in the comparison. However, the percentage setting may vary from language to language. Figure 3 shows that filtering 60% of original data still can improve the correlation performance for both to-English and from-English translation directions, confirming the robustness of this setting. For fitting most of the metrics and languages, we filter 60% of the instances out of the original test sets in the subsequent experiments.

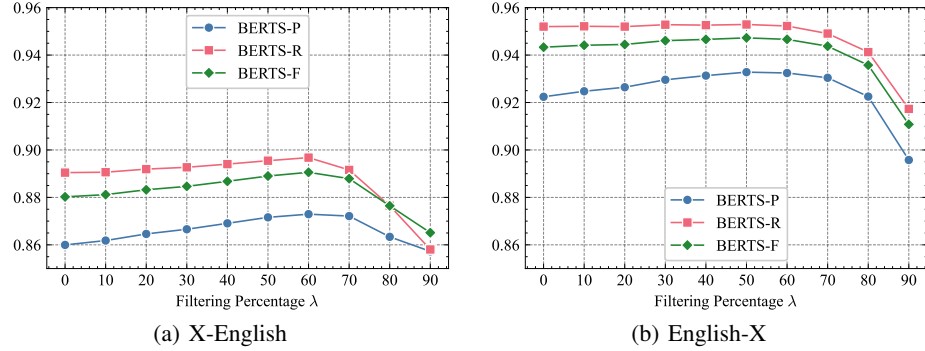

(a) X-English

(b) English-X

Figure 3: Comparison of averaged Pearson correlations measured on the WMT20 to/from English translation directions using different filtering percentages. Filtering 60% of original data works well for the two translation directions.

**Choice of the Filtering Metric** $\mathcal{M}$ matters because the discernment of a test instance can not be estimated without an accurate evaluation of the MT systems' performance. Figure 4 presents how the scores given by the different metrics affect the correlation of filtered test sets. Filtering the test set based on the scores given by the BERTS-R metric outperforms the test sets created by other metrics. It is reasonable that the BERTS-R metric consistently achieves the best correlation when using it as the evaluation metric (also as shown in Figure 2), and thus is better at quantifying the differences between the hypotheses and filtering out the non-discriminative instances. Although COMET is also remarkable in terms of Kendall's $\tau$ correlation, we chose to use the scores given by the BERTS-R metric as the bedrock to create the discriminative test set due to the balancing performance of these correlation coefficients.

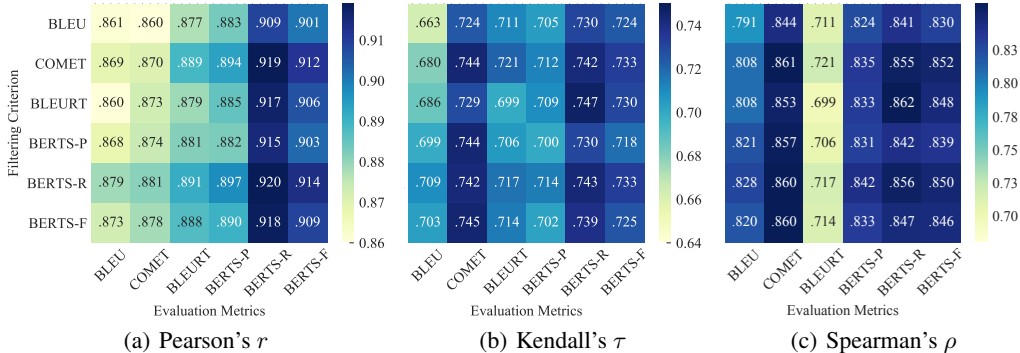

(a) Pearson's $r$

(b) Kendall's $\tau$

(c) Spearman's $\rho$

Figure 4: Comparison of averaged correlation results measured on all the WMT20 translation directions using different filtering metrics. Filtering the test sets by BERTS-R scores consistently yields stable correlation results across different evaluation metrics. Using COMET scores has comparable correlation performance with BERTR-S except Pearson correlation results.

## 4.3 Main Results

Using the filtering settings determined in the previous sections for other WMT benchmarks, Tables 2 and 3 present the comparison of correlation results between using the filtered and original test sets. The improved correlation performance across most metrics and benchmarks consistently confirms the greater effectiveness of evaluating with a variance-aware test set (VAT), especially for the metrics powered by pre-trained models. As for the $n$-gram-based metrics, it may over-penalize overlaps that share the same semantics, due to the hard-matching paradigm, making some VAT instances inactive in evaluating the diverse hypotheses. In contrast to the hard-matching paradigm, the metrics using the pre-trained models are able to fairly judge synonymous expressions, thus the created VAT are substantially useful to distinguish the MT systems.

Table 2: Comparison of averaged correlation results using original and variance-aware test sets (*VAT*) where Num denotes the number of language pairs. Evaluating MT systems with variance-aware test sets (*+VAT*) better correlates with human judgements across different evaluation metrics.

| Metric | WMT16 (Num=7) | | | WMT17 (Num=14) | | | WMT18 (Num=14) | | | WMT19 (Num=18) | | | WMT20 (Num=17) | | |
|---|---|---|---|---|---|---|---|---|---|---|---|---|---|---|---|
| | $\lvert r\rvert$ | $\lvert\tau\rvert$ | $\lvert\rho\rvert$ | $\lvert r\rvert$ | $\lvert\tau\rvert$ | $\lvert\rho\rvert$ | $\lvert r\rvert$ | $\lvert\tau\rvert$ | $\lvert\rho\rvert$ | $\lvert r\rvert$ | $\lvert\tau\rvert$ | $\lvert\rho\rvert$ | $\lvert r\rvert$ | $\lvert\tau\rvert$ | $\lvert\rho\rvert$ |
| BLEU | .826 | .645 | .778 | .910 | .737 | .865 | .827 | **.727** | **.802** | .912 | .762 | .878 | **.881** | .675 | .798 |
| *+VAT* | **.880** | **.723** | **.837** | **.928** | **.754** | **.876** | .827 | .723 | .796 | **.918** | **.786** | **.906** | .879 | **.709** | **.828** |
| COMET | .988 | **.886** | **.958** | .982 | .884 | .958 | .980 | **.925** | **.975** | .979 | **.882** | **.958** | .861 | .731 | .854 |
| *+VAT* | **.988** | .881 | .955 | **.985** | **.885** | **.959** | **.983** | .921 | .969 | **.981** | .862 | .948 | **.881** | **.743** | **.860** |
| BLEURT | .982 | .856 | .942 | .939 | .789 | .890 | .970 | **.900** | **.966** | .925 | .776 | .896 | .878 | .699 | .828 |
| *+VAT* | **.984** | **.859** | **.944** | **.951** | **.807** | **.906** | **.974** | .891 | .959 | **.935** | **.791** | **.902** | **.892** | **.717** | **.839** |
| BERTS-P | .970 | .848 | .924 | .951 | .806 | .909 | .965 | **.866** | **.949** | .953 | .811 | .911 | .886 | .699 | .827 |
| *+VAT* | **.976** | **.880** | **.948** | **.960** | **.820** | **.919** | **.978** | .865 | .949 | .953 | **.827** | **.924** | **.897** | **.714** | **.842** |
| BERTS-R | .941 | .831 | .931 | **.974** | .825 | **.926** | .915 | **.843** | .908 | **.961** | .821 | .924 | .916 | .742 | .853 |
| *+VAT* | **.953** | **.854** | **.943** | .972 | **.826** | .925 | **.953** | .842 | **.911** | .960 | **.834** | **.930** | **.920** | **.743** | **.856** |
| BERTS-F | .975 | .881 | .950 | .970 | .833 | .927 | .947 | .846 | .909 | **.963** | **.824** | .924 | .906 | .728 | .848 |
| *+VAT* | **.979** | **.900** | **.964** | **.974** | **.842** | **.929** | **.969** | **.873** | **.942** | .960 | .823 | **.925** | **.914** | **.733** | **.850** |

Table 3: Comparison of Pearson correlations using original and variance-aware test sets (*VAT*) on some mainstream language pairs. *T.* denotes the WMT test set. Using variance-aware test sets (*+VAT*) consistently improves the evaluation results of the language pairs across different test sets.

| Metric | De-En | | | En-De | | | Zh-En | | | En-Zh | | | En-Cs | | |
|---|---|---|---|---|---|---|---|---|---|---|---|---|---|---|---|
| | T.17 | T.18 | T.19 | T.17 | T.18 | T.19 | T.17 | T.18 | T.19 | T.17 | T.18 | T.19 | T.17 | T.18 | T.19 |
| BLEU | .928 | .969 | .888 | .819 | .980 | .952 | .869 | .983 | .900 | .980 | .947 | .902 | .956 | **.996** | .987 |
| *+VAT* | **.940** | **.975** | **.925** | **.845** | **.981** | .952 | **.894** | **.986** | .895 | **.980** | **.953** | **.925** | **.961** | .994 | **.994** |
| COMET | .989 | .997 | .947 | .935 | .989 | .987 | .979 | .988 | .989 | **.993** | .981 | .975 | .978 | .974 | .970 |
| *+VAT* | **.993** | **.998** | **.952** | **.950** | **.990** | **.993** | .979 | **.990** | **.992** | .990 | **.985** | **.976** | **.985** | **.976** | **.978** |
| BLEURT | .965 | .997 | .940 | .797 | .987 | **.982** | .915 | .984 | .984 | .797 | .883 | .807 | .919 | **.990** | **.987** |
| *+VAT* | **.979** | **.998** | **.944** | **.841** | **.987** | .981 | **.955** | **.988** | .984 | **.822** | **.914** | **.877** | **.947** | .986 | .984 |
| BERTS-P | .948 | .998 | .947 | .798 | .988 | .984 | .964 | .981 | .975 | .970 | .954 | .881 | .959 | .994 | .975 |
| *+VAT* | **.964** | **.999** | **.952** | **.830** | **.989** | **.989** | **.977** | **.984** | **.982** | **.982** | **.959** | **.926** | **.968** | **.998** | **.984** |
| BERTS-R | .988 | .997 | .946 | .909 | .990 | .991 | **.981** | .990 | .987 | **.994** | .976 | .940 | .982 | .997 | .984 |
| *+VAT* | **.989** | .997 | **.950** | **.912** | .990 | .991 | .978 | **.991** | .987 | .988 | **.980** | **.951** | **.984** | .997 | **.989** |
| BERTS-F | .973 | .999 | .949 | .859 | **.990** | .990 | .983 | .988 | .983 | .992 | .968 | .925 | .976 | .997 | .981 |
| *+VAT* | **.981** | **.999** | **.952** | **.876** | .989 | **.992** | **.988** | **.990** | **.986** | **.994** | **.972** | **.949** | **.979** | **.998** | **.987** |

## 4.4 Analysis of Variance-Aware Test Sets

To investigate how the correlation improvement benefits from VAT, we characterize the VAT built on the WMT20 benchmark from the perspective of their linguistic and data properties in this section.

**Sentence Length** generally associates with the translation difficulty (Koehn and Knowles, 2017), but the difficult sentence may be less relevant to the high discernment. As shown in Figure 5, longer sentences exhibit lower discernment and were filtered out by our method. Translating longer sentences is extremely challenging for MT systems due to the long-distance dependency or complex entity relationships (Cho et al., 2014; Sennrich and Haddow, 2016; Eriguchi et al., 2019), leading to close translation performance of MT systems. On the contrary, short sentences are more discriminative

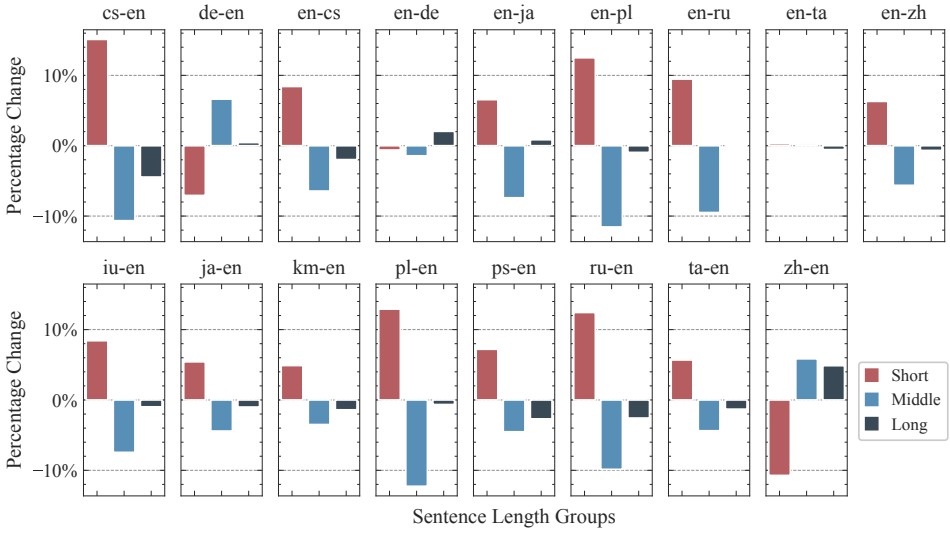

Figure 5: Absolute constitution changes of variance-aware test sets in terms of sentence length. VAT preserved more short sentences for most language pairs. The one-third of the sentences will be treated as the Long group, and the Medium/Short grouping methods are analogous.

because different systems tend to show greater differences in their syntactic and lexical choices. But some special translation directions show the opposite trends, such as Chinese→English and German→English. Not only is the number of systems that participated in these translation tasks relatively huge, but also the systems trained on these high-resource language pairs are likely to be more competitive. Since the competitive systems can be good at translating short sentences, the clues to judge their capability could rely on the translation of medium or long sentences.

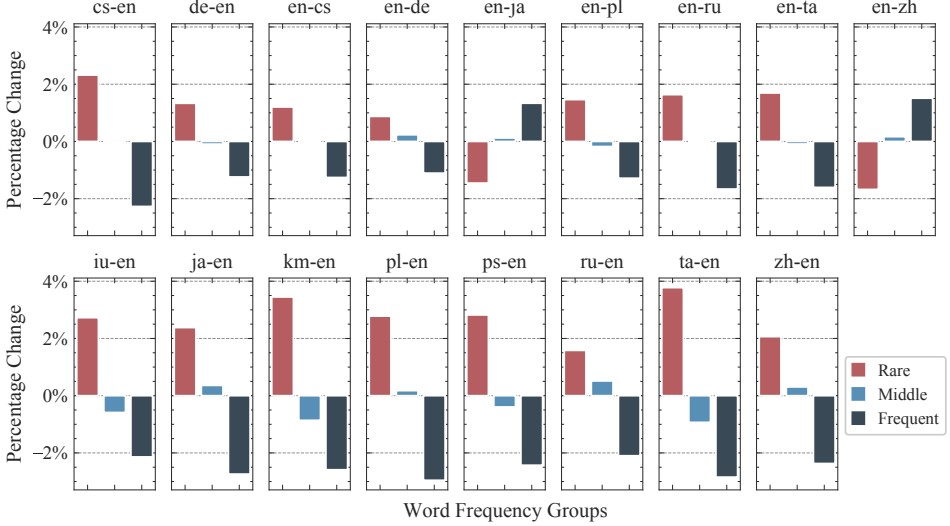

Figure 6: Absolute constitution changes of variance-aware test sets in terms of word frequency. VAT filtered the sentences which contain more frequent words. The boundaries for categorizing the "Rare", "Middle", "Frequent" group are 20%, 60%, 100% percentile of word frequency, respectively.

**Word Frequency** is a measure that reflects the finer-grained difference of MT systems since they may vary in their lexical choice of rare words (Koehn and Knowles, 2017; Ding et al., 2021b). As shown in Figure 6, the proportion of frequently occurring words in the training set is reduced in the VAT, indicating that high-frequency words are less discriminative. The representations of

high-frequency words learned on the training set tend to be stable, whereas the low-frequency words are insufficiently learned. In particular, some systems may enhance the translation performance of low-frequency words with the help of data augmentation (Fadaee et al., 2017; Ding et al., 2021a) or representation enhancement techniques (Nguyen and Chiang, 2018; Liu et al., 2019), resulting in the differences of lexical choice performed on the test set. Overall, the percentage change of word frequencies is not so large as it was for the comparison of sentence lengths, because the filtering operation is conducted at the sentence level, thus only those sentences whose proportion of low-frequency words is high will be preserved.

**Part-of-Speech** better depicts the lexical features of VAT considering the syntactic role a word plays. It can be seen from Figure 7 that VAT preserved more sentences containing proper nouns (NNP). This phenomenon echoes our previous comparative exploration of word frequency since there is a large overlap between NNPs and low-frequency words, such as the technical terms of a specific domain, but the translation performance on NNPs is not as intractable on long sentences. Due to the fact that the bottleneck of long-sentence translation may be related to the model architecture (Cho et al., 2014), most MT systems that share a homogeneous architecture (Vaswani et al., 2017) still have problems in translating these challenging sentences. Similar to the problem of low-frequency words, the poor translation accuracy for NNPs can be alleviated by introducing external knowledge (Chatterjee et al., 2017) or domain adaptation techniques (Hu et al., 2019) concerning data-efficient learning, thus evaluating the translation of NNPs is also valuable in distinguishing MT systems.

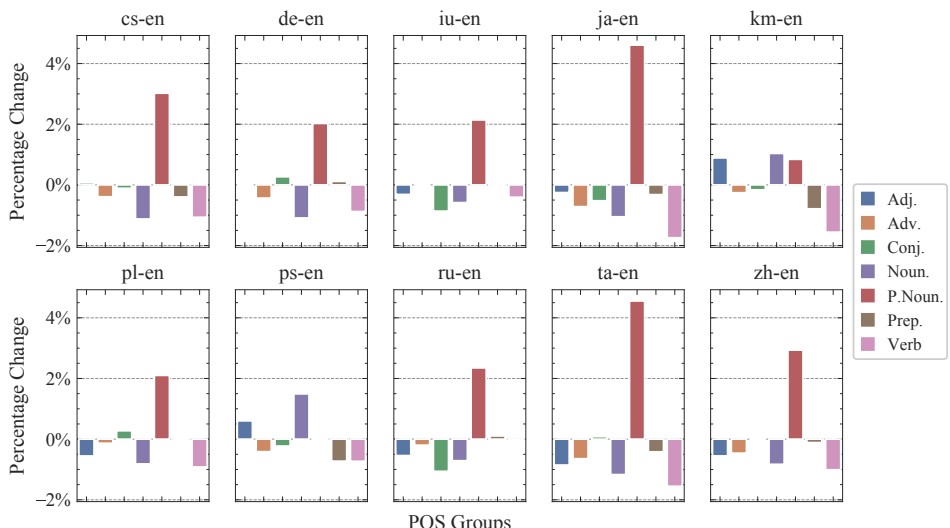

Figure 7: Absolute constitution changes of variance-aware test sets in terms of English part-of-speech tagged by the NLTK toolkit. VAT has more proper nouns than the original test sets.

**Human Paraphrasing** is another option for optimizing existing MT test sets. It asks human experts to paraphrase the references as much as possible (Freitag et al., 2020b). This can effectively improve the correlation of automatic metrics but is very costly. In this experiment, we investigate the relationship between human paraphrased test sets provided by Freitag et al. (2020b) and the preserved (discriminative) and filtered out (non-discriminative) subsets by our approach.

Table 4 gives the averaged edit distance (Levenshtein, 1966): a large value means more paraphrases on a subset. The results show that human experts need to produce more paraphrases for the filtered out test instances whereas making fewer paraphrases for the preserved ones. This means that the preserved subsets are of higher quality for MT evaluation, and thus can improve the correlation with human judgements.

Table 4: Edit distances between different subsets of the English→German test set with corresponding human paraphrased data. Human experts need to do fewer paraphrases on the preserved subsets.

|         | **All** | **Filtered Out** | **Preserved** |
|---------|---------|------------------|---------------|
| **WMT20** | 30.35   | 30.46            | **30.18**     |
| **WMT19** | 19.82   | 20.25            | **19.17**     |
| **WMT18** | 20.02   | 20.30            | **19.72**     |

**Translation Difficulty**   is not the same as the evaluation discernment as we assumed before. The test instances whose difficulty lies at the extremes of the scale are not useful in an evaluation aiming to distinguish between MT systems, so the preserved samples possibly are not the simplest or the most difficult instances. Starting from this intuition, we investigate the distribution of the averaged score of the test instances on the WMT20 English→German translation task since it involves competitive systems that are challenging for the automated evaluation (Freitag et al., 2020b). Obviously, Figure 8 reveals that the preserved instances have moderate but not extreme difficulty. The phenomenon that samples with slightly higher difficulty are preserved also conforms with the previous observation in terms of the sentence length, that the longer sentences are more vital in distinguishing the MT systems due to their strong capability, also echoes the previous research stating that this translation direction easily confused the automatic evaluation metric (Bojar et al., 2018; Barrault et al., 2019).

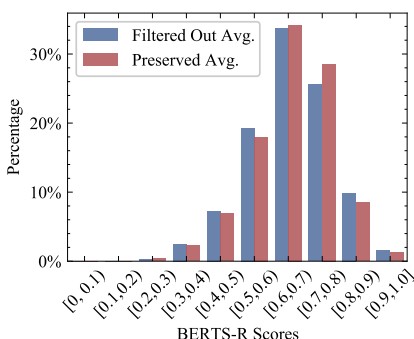

Figure 8: Comparison of distribution of BERTS-R scores between filtered out and preserved sentences. Sentences with medium difficulty are more discriminative than which are either extremely hard or extremely easy.

To conclude, a test item preserved by the proposed filtering method is discriminative in terms of its linguistic and data properties, thus the improvement in the correlation of the variance-aware test set is reasonable. Moreover, the variance-aware filtering method has the potential for saving the human labor for diversifying the test sets.

## 5   Conclusions and Future Work

This paper introduces a method to select discriminative test instances from the machine translation benchmark and automatically create a series of variance-aware test sets. Experimental results show that using the created test sets can improve the correlation performance of automatic evaluation results across representative test sets and languages, confirming the effectiveness and generality of the proposed method. Further analysis of the features of the test instances supports the rationality of variance-aware test sets and ensures its reliability for other possible uses.

Future work includes: 1) investigating the use of the variance-aware test sets in other MT research questions. For example, using them as the validation sets in some time-consuming scenarios, e.g., neural architecture search and reinforcement learning; 2) applying the filtering method to the training set to accelerate the learning process; 3) extending the filtering method to other evaluation tasks like dialogue generation.

## Acknowledgments

This work was supported in part by the National Natural Science Foundation of China (Grant No. 61672555), the Science and Technology Development Fund, Macau SAR (Grant No. 0101/2019/A2), and the Multi-year Research Grant from the University of Macau (Grant No. MYRG2020-00054-FST). We thank the anonymous reviewers for their insightful comments.

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
