# OpenReview forum: "Variance-Aware Machine Translation Test Sets"
_NeurIPS.cc/2021/Track/Datasets_and_Benchmarks/Round1 — NeurIPS 2021 Datasets and Benchmarks Track (Round 1)_

### Official Review · Reviewer_6NR4 · 2021-06-20
**A simple, nice way of creating compact and useful evaluation datasets**

**Rating:** 8
**Confidence:** 3

**Strengths:**

While the technique is simple, I think it is an interesting approach, and correlation with human evaluations shows that the resulting test sets meet their intended purpose of providing a better evaluation while being smaller. Thus, I think this contribution will have a meaningful impact and will be used.

The presentation and analysis is rather thorough: several MT evaluation metrics are considered (which also serves, indirectly, as a comparison between metrics), as well as several correlation coefficients, apart from the hyperparameter search for the optimal variance percentile threshold for each metric.

The paper not only provides the technique and its evaluation, but also some insights on the kind of instances that the system is keeping and removing.

**Weaknesses:**

It would have been nice to have some language-specific analysis in Sections 4.2 and 4.3 (e.g. does the technique work better for some language pairs than others?) apart from 4.4, although I understand it can be difficult for space reasons.

**Additional Feedback:**

I confirm that I have read the author rebuttal. It has reaffirmed my positive view of the paper.

Why does BLEURT behave so differently with respect to Spearman's rho (in Figure 3c) with respect to the other two correlation coefficients? It achieves good results with Pearson and Kendall but rather poor ones in terms of Spearman's rho. I think this merits a comment.

Figure 7 would be easier to see if the bars were non-overlapping (using side-by-side bars of different colors).

l. 19: metrics that exploits -> metrics that exploit

l. 73: do not consider semantic -> do not consider semantics

l. 76: making evaluate the semantic overlap possible -> making it possible to evaluate the semantic overlap

l. 104: paraphrased references have been proved that it can significantly improve -> paraphrased references have proved that they can significantly improve

l. 120: a discriminative test instance must make the evaluation exhibits -> a discriminative test instance must make the evaluation exhibit

l. 144: since it is the well-recognized benchmarks -> since they are the well-recognized benchmarks

l. 181: the metrics driving by the pre-trained models -> the metrics driven by the pre-trained models

l. 219: are like to be -> are likely to be

**Clarity:**

The paper is well written and clear, except for the issues with the "Human paraphrasing" part mentioned above.

Several references are made to "the entrance examination" which I suppose refers to the Chinese entrance examination (gaokao). Perhaps this should be specifically mentioned for the sake of readers from other cultures which may not find the reference obvious (or another option is to just make the analogy with exams in general, as what is said is arguably true of any exam).

The paper would probably benefit from some proof-reading - I found a bunch of typos and grammar issues (listed in the "Additional Feedback" below) but I may have missed more, although there's nothing that affects the understandability of the content.

**Correctness:**

The experimental design is thorough and well explained, experiments are extensive and the claims are sound as far as I can see, with one small example: I am not sure I totally understand the conclusions of the "Human paraphrasing" analysis in Page 9. If I understand correctly, what Table 4 says is that the method tends to reserve the sentences from the original test set that are more similar to the paraphrases. But where does the conclusion that "deeper reason of correlation improvement brought by VAT may be similar to the effect of human paraphrasing" come from? Can't this table be explained, for example, by the fact that for more discriminative examples, humans generate closer paraphrases (be it due to their length, the kind/frequency of words they contain, etc.) In my view the claim of that conclusion is not supported, although it might be that I'm missing some point of the analysis and it can be fixed by some clarification or rephrasing.

The rest of the analyses all look sound and clear to me.

**Documentation:**

The datasets are adequately documented and licensed.

**Ethics:**

The resource relies on text from existing datasets which are widely used in the community (from which some sentences are filtered) so I do not see any ethical concerns.

**Relation To Prior Work:**

The paper is adequately contextualized with respect to prior work.

**Summary And Contributions:**

The paper presents a technique to filter machine translation test sets in order to keep a set of instances that allow to better evaluate and compare systems, as well as a set of test sets (for 70 language pairs) generated using this technique. The main idea is straightforward: to keep those test instances where the performance of different systems (as measured by one of a set of possible metrics) has a large variance, and filter away those where the performance is very similar (be it because they are very easy, and all the systems get them right, or very fall such that they all fail). The resulting test sets can be evaluated and compared to the original test sets in terms of correlation to human judgments. The authors perform such an evaluation using a range of different MT evaluation metrics, variance thresholds and correlation coefficients, concluding that correlation to human evaluation typically increases with respect to the original test sets (i.e., the smaller filtered test sets provide a better evaluation of systems, at least measured against the human standard). They also include some analysis of the kept vs. filtered instances in terms of characteristics like sentence length, PoS tags or translation difficulty, which provides some insight on what the technique is doing.

---

> ### Author Response · Authors · 2021-07-12
> **Response to Reviewer 6NR4**
>
> Thanks for the review.
>
> > **Q1** *It would have been nice to have some language-specific analysis in Sections 4.2 and 4.3 (e.g. does the technique work better for some language pairs than others?) apart from 4.4, although I understand it can be difficult for space reasons.*
>
> Thank you for the suggestion. The degree of correlation improvement varies from language pairs to language pairs. We will compare the performance of different languages in terms of the experiment in Section 4.2/4.3, and give language-specific results and analysis in the revised Appendix. This might provide some insights for future research.
>
> >  **Q2**  *I am not sure I totally understand the conclusions of the "Human paraphrasing" analysis in Page 9. If I understand correctly, what Table 4 says is that the method tends to reserve the sentences from the original test set that are more similar to the paraphrases. But where does the conclusion that "deeper reason of correlation improvement brought by VAT may be similar to the effect of human paraphrasing" come from? Can't this table be explained, for example, by the fact that for more discriminative examples, humans generate closer paraphrases (be it due to their length, the kind/frequency of words they contain, etc.) In my view the claim of that conclusion is not supported, although it might be that I'm missing some point of the analysis and it can be fixed by some clarification or rephrasing.*
>
> Sorry for the misunderstanding. In the revised version, we have rephrased the related paragraph to make the claim more clear. The result reveals that human experts need to do more paraphrases for the filtered test instances whereas to do fewer paraphrases for the reserved ones.
>
> > **Q3** *Several references are made to "the entrance examination" which I suppose refers to the Chinese entrance examination (gaokao). Perhaps this should be specifically mentioned for the sake of readers from other cultures which may not find the reference obvious (or another option is to just make the analogy with exams in general, as what is said is arguably true of any exam).*
>
> Thanks for your constructive suggestions. We agree that the conception of “the entrance examination” may be unclear for readers with different cultures. We have updated the related text in the revised version.
>
> > **Q4** *Why does BLEURT behave so differently with respect to Spearman's rho (in Figure 3c) with respect to the other two correlation coefficients?*
>
> BLEURT worked fine considering the Pearson’s and Kendall’s correlation coefficients. According to the reliable research papers, Spearman's rho sometimes is a "harsh" correlation coefficient [1] in terms of MT meta-evaluation and it is also no longer used in the recent research [2,3]. It may over-penalize the wrong ranking of two similar systems where even humans are uncertain to distinguish [1,2,3]. We use Spearman's rho because we want to cover the meta-evaluation metrics as much as possible.
>
> Reference:
>
> [1] Macháček, M., & Bojar, O. (2013). Results of the WMT13 metrics shared task. In Proceedings of the Eighth Workshop on Statistical Machine Translation (pp. 45-51). https://aclanthology.org/W13-2202.pdf
>
> [2] Macháček, M., & Bojar, O. (2014). Results of the WMT14 metrics shared task. In Proceedings of the Ninth Workshop on Statistical Machine Translation (pp. 293-301). https://aclanthology.org/W14-3336.pdf
>
> [3] Mathur, N., Baldwin, T., & Cohn, T. (2020). Tangled up in BLEU: Reevaluating the Evaluation of Automatic Machine Translation Evaluation Metrics. In Proceedings of the 58th Annual Meeting of the Association for Computational Linguistics (pp. 4984-4997). https://aclanthology.org/2020.acl-main.448.pdf
>
> > **Q5** *Regarding the typos and the presentation style of Figure 7.*
>
> Thanks for your constructive comments. We have fixed the typos and updated Figure 7 in the revised version.

---

> > ### Comment · Reviewer_6NR4 · 2021-07-12
> > **Thanks for the answers**
> >
> > Thanks for the answers, I think they satisfactorily clarify the (minor) unclear points I raised, and reaffirm my positive view of the paper.

---

### Official Review · Reviewer_rNkG · 2021-07-02
**A great addition to machine translation evaluation toolkit**

**Rating:** 8
**Confidence:** 3

**Strengths:**

The paper is well written with a clear structure. The paper provides a novel method of automatically generating high-quality test sets for machine translation evaluation and publishes the generated test sets to promote reproducibility and to allow researchers to use the generated test sets for evaluating machine translation models. The authors also promise to release the code for automatic test set generation which will be a highly valuable asset for future research.

**Weaknesses:**

In general there is not many weaknesses in the paper. The only section that I found a bit lacking was the 3.1 Motivation, which describes the motivation behind variance and variance-aware test sets. I am not sure the example from entrance examinations is the best one. I would propose rewriting the section focusing on why variance-awareness is important in machine translation in particular.

**Additional Feedback:**

Good work! I look forward to seeing this paper published.

**Clarity:**

The paper is clear and well written for the most part. As stated above, I would suggest rewriting the section 3.1 Motivation.

**Correctness:**

Based on the information provided in the paper and without seeing the actual code, the datasets seem to be constructed in a sound way. Review of the source code would be required to make sure the implementation is correct.

**Documentation:**

Dataset generation is documented well, apart from not making the source code available for reviewers.

**Ethics:**

As the datasets are generated by filtering existing datasets, I do not see any new ethical issues arising. However, I am not an expert on ethics.

**Relation To Prior Work:**

The discussion of related work is detailed with good references to previous work.

**Summary And Contributions:**

The paper introduces automatically generated, variance-aware test sets for machine translation, including 70 test sets covering 35 translation directions based on the datasets from WMT16 to WMT20.

The authors first describe the problems of machine translation evaluation and highlight the shortcomings of the currently available approaches. They then describe the concept of variance-awareness and explain the novel filtering methodology to generate the test sets.

The paper makes a significant contribution to the field of machine translation evaluation by not only providing the datasets but providing a automated methodology for generating variance-aware tests sets from different datasets and providing detailed analysis of discriminative and indiscriminative test instance.

---

> ### Author Response · Authors · 2021-07-12
> **Response to Reviewer rNkG**
>
> Thanks for the review.
>
> > **Q1** *The only section that I found a bit lacking was the 3.1 Motivation, which describes the motivation behind variance and variance-aware test sets. I am not sure the example from entrance examinations is the best one. I would propose rewriting the section focusing on why variance-awareness is important in machine translation in particular.*
>
> Thank you for the suggestion. We have followed your suggestion to update this part and focus on the MT evaluation case to illustrate our idea. We simply start from telling the facts in the general tests/exams, and focus on MT evaluation by making the analogy with the real-world facts. Finally, use the three cases in Figure 1 to echo the motivation.
>
> > **Q2** *Review of the source code would be required to make sure the implementation is correct.*
>
> We have released an early version of the source code (https://github.com/NLP2CT/Variance-Aware-MT-Test-Sets) which includes the data and implementation of generating the VAT. We will keep updating the code and related documentation after the response.

---

### Official Review · Reviewer_AV8U · 2021-07-03
**A method to preserve only discriminative sentences in MT datasets**

**Rating:** 7
**Confidence:** 2
**Correctness:** Yes, the main claim of the paper is c…

**Strengths:**

1.  The premise and motivation of the work is valid, IMO. If the community designs a dataset, for MT or any other NLP task, where all the models perform very close, then these sentences are not really useful to score and discriminate among systems. We should rather focus on the sentences that actually can discriminate among systems. According to the authors (lines 32-33), previous attempts are based on human and automatic paraphrasing and are human or computational expensive. On the contrary, the authors method seems intuitive and it’s inspired by simple statistics, arguing that output showing high score variance (those showing more variance than x% of the test samples) are the ones that serve to discriminate the systems.

**Weaknesses:**

1. The authors show that their technique improves correlations to human judgements. However, I feel the authors also would need to show whether the reference-based metrics and system ranking change or not on that subset, to check whether it would be actually crucial to consider their technique in MT standard evaluations.

2. In the contributions (lines 49-58), the authors mention that their approach reduces high computational costs, since it reduces the size of the test set for RL and NMT systems. Yet, I expect the bottleneck of the computational cost is on the training set, as the test set should be executed (ideally) just once. In this context, I wonder how the approach truly contributes to ‘green AI’ as all sentences are used during training. Note that the test set is just a representation of the real world, where the model will be deployed and translate huge amounts of sentences, including very easy and very difficult ones. Would it be possible to expand on this?

3. I see how the methods can be used to present variants of MT datasets and benchmarks, but the paper itself does not present one. Although I don’t see this as a major weakness, I also feel the fit with this track might not be perfect.


**Additional Feedback:**

1. Is ‘lightweight’ (line 1) used here as a synonym of ‘small’?
2. Could you specify what you exactly mean by BERT-family models in line 167?
3. What are the MT models used to compute the outputs? I might have missed it, but I only can see a mention of ‘system hypotheses’ in line 150.
4. The goal of the paper is to obtain the samples so that the translations for those samples show a higher correlation with respect to the human judgements in comparison to the whole test set, which the authors achieve. However, what if we simply remove references where the scores by the systems are very low, wouldn’t we potentially obtain a higher correlation? Although easy sentences might be useless to discriminate among systems, I presume they would help to obtain even a higher correlation which is what this paper shows, or I might be missing something? I’m not saying it’s something it should be done, but could it be something that happens?
5. Could Figure 4 specify what is considered a short, medium or long sentence?
6. The human paraphrasing paragraph (lines 245-259) was not clear to me, if the paper is accepted, would it be possible to detail it a bit more?


**Clarity:**

The paper is overall well-written. I have some minor questions to the authors that I stated in the additional feedback.

**Documentation:**

The code to compute the variants of the MT test sets is available and clearly specified in the abstract.

**Ethics:**

I don’t see major ethics concerns with this work.

**Relation To Prior Work:**

Yes. The authors argue that the main difference is that previous efforts focused on increasing the reference diversity through human and automatic paraphrasing, but that these methods are expensive.

**Summary And Contributions:**

This paper proposes a method to select reference samples from a MT dataset that are useful to discriminate among the output of MT systems, aiming to reserve sentences for which MT systems produce different/diverse outputs.

The idea of the paper is that given X metrics that are used to evaluate the models, discriminative reference samples must be those that cause the systems to score very differently among them. This essentially translates into very easy and difficult sentences being little discriminative, as all the system will obtain very good or bad scores, respectively.

The authors show that their variance-aware subsets obtain higher correlations with respect to human judgements than the original datasets (WMT16 to WMT20).

---

> ### Author Response · Authors · 2021-07-12
> **Response to Reviewer AV8U (Part 1)**
>
> Thanks for the review.
>
> > **Q1** *I feel the authors also would need to show whether the reference-based metrics and system ranking change or not on that subset, to check whether it would be actually crucial to consider their technique in MT standard evaluations.*
>
> We entirely agree that the change of system ranking is an important factor for verifying the effectiveness of the introduced test set. Therefore, besides reporting the widely-used Pearson’s correlation which is based on the overall distribution of evaluation scores, we also report Spearman’s and Kendall’s correlations whose calculations are based on the **overall system ranking**.  A higher Spearman’s or Kendall’s correlation means a better system ranking, and the improvements of Spearman’s and Kendall’s correlations have been reflected in Tables 2 and 3 of the paper.
> Furthermore, we also give a case study of the competitive WMT20 Chinese-English translation task as shown in Table I, the result shows that the introduced test set indeed provides a better ranking. As shown in the following table, the BERTScore gives the same ranking as human evaluators on VAT but fails on the original test sets.
>
> *Table I. A case of ranking given by the human and automated evaluation*
>
> | SYSTEM                   | HUMAN Top4 | BERTS-F Top4  | BERTS-F +VAT  Top4 |
> | :----------------------- | :--------: | :-----------: | :----------------: |
> | Huoshan_Translate.919    | 0.1020 (1) | 0.6873 (3, ×) |   0.6854 (1, √)    |
> | DiDi_NLP.401             | 0.0887 (2) | 0.6840 (4, ×) |   0.6821 (2, √)    |
> | WeChat_AI.1525           | 0.0772 (3) | 0.6875 (1, ×) |   0.6813 (3, √)    |
> | Tencent_Translation.1249 | 0.0631 (4) | 0.6874 (2, ×) |   0.6812 (4, √)    |
>
>
>
> > **Q2** *In the contributions (lines 49-58), the authors mention that their approach reduces high computational costs, since it reduces the size of the test set for RL and NMT systems. Yet, I expect the bottleneck of the computational cost is on the training set, as the test set should be executed (ideally) just once. In this context, I wonder how the approach truly contributes to ‘green AI’ as all sentences are used during training. Note that the test set is just a representation of the real world, where the model will be deployed and translate huge amounts of sentences, including very easy and very difficult ones. Would it be possible to expand on this?*
>
> Our proposed test sets have a positive impact on "Green AI":
>
> 1. For most MT research, the WMT test sets are not only used as the test set but also the **validation** **set** that needs to be executed multiple times (about >30 times). For each validation during model training, the model needs to re-compute the perplexity or BLEU scores of the validation set, which is time-consuming. Our proposed test set is much smaller than the original one, which can significantly reduce the validation time. Furthermore, the improved correlations to human judgments might enable the model to find the final optima early and thus perform an earlier early stopping, saving the overall training time.
>
> 2. The size of the validation set has a greater impact on the research lines of reinforcement learning and neural architecture search for MT. For example, the Evolved Transformer [1] searches 6000 model architectures with 446M training steps, and thus the validation set will be used 446M/[validation interval] times. A smaller validation set can no doubt save much time. Therefore, we think that our proposed test sets can also bring benefits to these research lines of MT.
>
>    Reference:
>
>    [1] So, D., Le, Q., & Liang, C. (2019). The evolved transformer. In the International Conference on Machine Learning (pp. 5877-5886). PMLR. http://proceedings.mlr.press/v97/so19a/so19a.pdf
>
> 3. Inspired by the comment that "**expect the bottleneck of the computational cost is on the training set**", we find that our proposed variance-aware filter method might also be used to the training set by employing different MT systems to translate the training set and filtering those indiscriminative and hard-to-translate training instances (like the first example in Figure 1). The filtered training set might be used to train a better MT system with less time. We will put it as our future works and have clarified it in the conclusion of the revised version. Thank you again for your comment.

---

> > ### Author Response · Authors · 2021-07-12
> > **Response to Reviewer AV8U (Part 2)**
> >
> > > **Q3** *I see how the methods can be used to present variants of MT datasets and benchmarks, but the paper itself does not present one. Although I don’t see this as a major weakness, I also feel the fit with this track might not be perfect.*
> >
> > As clarified in the call for papers of the datasets and benchmarks (https://nips.cc/Conferences/2021/CallForDatasetsBenchmarks), the scope of this track is *“In addition to new datasets and benchmarks on new or existing datasets, we welcome submissions that detail advanced practices in data collection and curation that are of general interest even if the data itself cannot be shared. Data generators or reinforcement learning environments are also in scope. Frameworks for responsible dataset development, audits of existing datasets, **identifying significant problems with existing datasets and their use**, or systematic analyses of existing systems on novel datasets that yield important new insight are also in scope.”*
> >
> > In this paper, we identify significant problems (i.e., containing indiscriminative test instances) with existing datasets (i.e., the widely-used WMT test sets), and further provide a solution (i.e., variance-aware filtering) and some insights for future research (i.e., analysis of discriminative and indiscriminative test instance). The WMT test sets contribute a lot to the community, for example, both the papers of attention mechanism [2] (>18,000 citations) and Transformer [3] (>23,000 citations) verify their methods on the WMT test sets. Therefore, we think that our paper is well fit for NeurIPS 2021 Datasets and Benchmarks Track.
> >
> > Reference:
> >
> > [2] Bahdanau, D., Cho, K., & Bengio, Y. (2014). Neural machine translation by jointly learning to align and translate. arXiv preprint arXiv:1409.0473. https://arxiv.org/abs/1409.0473
> >
> > [3] Vaswani, A., Shazeer, N., Parmar, N., Uszkoreit, J., Jones, L., Gomez, A.N., Kaiser, Ł. and Polosukhin, I., (2017). Attention is all you need. In Advances in neural information processing systems (pp. 5998-6008). https://arxiv.org/abs/1706.03762
> >
> > > **Q4** *Is ‘lightweight’ (line 1) used here as a synonym of ‘small’?*
> >
> > Yes. We have replaced "lightweight" with "small" in the revised version.
> >
> > > **Q5** *Could you specify what you exactly mean by BERT-family models in line 167?*
> >
> > The BERT-family models used in this paper follow the official Implementation of BERTScore (https://github.com/Tiiiger/bert_score#default-model), e.g., *RoBERTa-large* for evaluating English text and *BERT-base-multilingual-case* for evaluating German and French text. We have clarified this in the revised version.
> >
> > > **Q6** *What are the MT models used to compute the outputs? I might have missed it, but I only can see a mention of ‘system hypotheses’ in line 150.*
> >
> > The MT models are the competitive MT models participating in the WMT competitions. Take the WMT20 Chinese-English translation task as an example, there are 16 submitted results from the participating translation systems (models). Our proposed method calculates the score variance of these outputs and generates a novel variance-aware test set. All the related data can be found on the WMT official websites each year (http://www.statmt.org/wmt20/results.html). We will clarify this part in the revised Appendix.

---

> > > ### Author Response · Authors · 2021-07-12
> > > **Response to Reviewer AV8U (Part 3)**
> > >
> > > > **Q7** *What if we simply remove references where the scores by the systems are very low, wouldn’t we potentially obtain a higher correlation? Although easy sentences might be useless to discriminate among systems, I presume they would help to obtain even a higher correlation which is what this paper shows, or I might be missing something? I’m not saying it’s something it should be done, but could it be something that happens?*
> > >
> > > We follow your suggestion to remove the references where the averaged scores of the systems are very low, and construct a new test set called difficulty-aware test set (DAT). We follow the experiment procedure of the paper to explore the choice of filtering percentage. We compare the correlation of DAT and baseline (original test set) under different settings, the results tested on WMT20 as shown in Tables II, III, IV.
> > >
> > > *Table II. Comparison of averaged Pearson's correlations of DAT.*
> > >
> > > |     Filter % |   BLEU    |   COMET   |  BLEURT   |  BERTS-P  |  BERTS-R  |  BERTS-F  |
> > > | -----------: | :-------: | :-------: | :-------: | :-------: | :-------: | :-------: |
> > > | 0 (Baseline) | **0.881** | **0.861** |   0.878   | **0.886** | **0.916** | **0.906** |
> > > |           10 |   0.880   |   0.859   |   0.876   | **0.886** |   0.915   |   0.905   |
> > > |           20 | **0.881** |   0.857   |   0.875   |   0.884   |   0.912   |   0.903   |
> > > |           30 |   0.880   |   0.854   |   0.876   |   0.884   |   0.912   |   0.903   |
> > > |           40 |   0.880   |   0.852   |   0.876   |   0.882   |   0.911   |   0.901   |
> > > |           50 |   0.880   |   0.850   |   0.877   |   0.883   |   0.908   |   0.901   |
> > > |           60 |   0.879   |   0.847   | **0.879** |   0.881   |   0.907   |   0.899   |
> > > |           70 |   0.872   |   0.843   |   0.877   |   0.879   |   0.905   |   0.897   |
> > > |           80 |   0.867   |   0.832   |   0.874   |   0.875   |   0.897   |   0.891   |
> > > |           90 |   0.856   |   0.806   |   0.861   |   0.860   |   0.871   |   0.870   |
> > >
> > > *Table III. Comparison of averaged Kendall's correlations of DAT.*
> > >
> > > |     Filter % |   BLEU    |   COMET   |  BLEURT   |  BERTS-P  |  BERTS-R  |  BERTS-F  |
> > > | -----------: | :-------: | :-------: | :-------: | :-------: | :-------: | :-------: |
> > > | 0 (Baseline) |   0.675   |   0.731   |   0.699   |   0.699   | **0.742** | **0.728** |
> > > |           10 |   0.673   | **0.735** |   0.703   |   0.699   |   0.726   |   0.727   |
> > > |           20 |   0.679   |   0.727   |   0.691   |   0.699   |   0.720   |   0.719   |
> > > |           30 | **0.683** |   0.719   |   0.704   | **0.704** |   0.730   |   0.717   |
> > > |           40 |   0.682   |   0.731   |   0.711   |   0.697   |   0.727   |   0.705   |
> > > |           50 |   0.673   |   0.732   |   0.706   |   0.693   |   0.737   |   0.718   |
> > > |           60 |   0.678   |   0.720   | **0.712** |   0.698   |   0.721   |   0.724   |
> > > |           70 |   0.670   |   0.704   |   0.701   |   0.696   |   0.712   |   0.717   |
> > > |           80 |   0.641   |   0.725   |   0.687   |   0.674   |   0.709   |   0.705   |
> > > |           90 |   0.618   |   0.685   |   0.662   |   0.654   |   0.652   |   0.662   |
> > >
> > > *Table IV. Comparison of averaged Spearman's correlations of DAT.*
> > >
> > > |     Filter % |   BLEU    |   COMET   |  BLEURT   |  BERTS-P  |  BERTS-R  |  BERTS-F  |
> > > | -----------: | :-------: | :-------: | :-------: | :-------: | :-------: | :-------: |
> > > | 0 (Baseline) |   0.798   | **0.854** | **0.828** |   0.827   | **0.853** | **0.848** |
> > > |           10 |   0.797   |   0.852   |   0.703   |   0.823   |   0.841   |   0.842   |
> > > |           20 |   0.799   |   0.849   |   0.691   |   0.818   |   0.839   |   0.835   |
> > > |           30 | **0.811** |   0.842   |   0.704   | **0.828** |   0.844   |   0.833   |
> > > |           40 | **0.811** |   0.851   |   0.711   |   0.820   |   0.842   |   0.826   |
> > > |           50 |   0.804   |   0.853   |   0.706   |   0.815   |   0.849   |   0.833   |
> > > |           60 | **0.811** |   0.842   |   0.712   |   0.818   |   0.838   |   0.841   |
> > > |           70 |   0.802   |   0.832   |   0.701   |   0.816   |   0.834   |   0.834   |
> > > |           80 |   0.779   |   0.840   |   0.687   |   0.804   |   0.837   |   0.835   |
> > > |           90 |   0.766   |   0.805   |   0.662   |   0.794   |   0.793   |   0.799   |
> > >
> > > The results show that DAT does not improve the correlations compared to the baseline in most cases. But VAT can gain improvements as presented in Tables 2&3 of the paper.
> > >
> > > > **Q8** *Could Figure 4 specify what is considered a short, medium or long sentence?*
> > >
> > > The first $\frac{1}{3}$ longest sentences will be treated as the Long group, and the Medium/Short grouping methods are analogous. We have clarified this in the revised version.
> > >
> > > > **Q9** *The human paraphrasing paragraph (lines 245-259) was not clear to me, if the paper is accepted, would it be possible to detail it a bit more?*
> > >
> > > Sorry for the misunderstanding. We have updated this paragraph in the revised version and will keep updating it after the discussion.

---

> > > > ### Comment · Reviewer_AV8U · 2021-07-13
> > > > **Response to Authors**
> > > >
> > > > Thank you very much for this very detailed author response(s). Overall, I am more convinced about the paper with respect to its fit, the overall ranking concerns, and experiments regarding Q7, and consequently raised my score a bit.
> > > >
> > > > I am not fully convinced about the 'green AI' part. I understand the (updated) motivation that includes the training set, but in the current version it's not possible to conclude that this will happen (I have to confess that I am relatively optimistic about it, but there are no 'numbers' and I cannot  evaluate this). If the paper is accepted and the authors can do something regarding this point, that would be great.

---

### Comment · Area_Chair_mQvR · 2021-07-19
**Quick question**

Hi Authors,

Thanks for the thoughtful replies to reviewers' comments!
I have a quick question regarding this paper, following up on a question by a reviewer:

------
Q6 What are the MT models used to compute the outputs? I might have missed it, but I only can see a mention of ‘system hypotheses’ in line 150.
Answer: The MT models are the competitive MT models participating in the WMT competitions. Take the WMT20 Chinese-English translation task as an example, there are 16 submitted results from the participating translation systems (models). Our proposed method calculates the score variance of these outputs and generates a novel variance-aware test set. All the related data can be found on the WMT official websites each year (http://www.statmt.org/wmt20/results.html). We will clarify this part in the revised Appendix.
-----

Is it correct to interpret that the models used to compute variances (and thus, used to filter the evaluation set) are also the models whose outputs are evaluated? If so, how would the performance vary if you use _different_ sets of models to compute variances? A realistic scenario would be (1) creating a variance-aware filtered evaluation set with some existing systems and then (2) using the new evaluation set to evaluate _new_ systems.  Please feel free to correct me if I'm missing something, thanks!

---

> ### Public Comment · ~Hou_Pong_Chan2 · 2021-07-20
> **Response to Area Chair mQvR**
>
> Thanks for the questions. OpenReview has denied the permission of the authors to reply to comments, thus we have to answer your questions by our colleague's account.
>
> >**Q1** *Is it correct to interpret that the models used to compute variances (and thus, used to filter the evaluation set) are also the models whose outputs are evaluated?*
>
> Yes, the interpretation is correct. The results show that this can increase the correlation to human judgments and provide a better system ranking (an example is given in Table I of the response to Reviewer AV8U). In the future evaluation of MT systems, researchers can follow this way (i.e., the models used to compute variances are also the models whose outputs are evaluated) to accurately evaluate the systems and find which system is the best.
>
> >**Q2** *If so, how would the performance vary if you use different sets of models to compute variances? A realistic scenario would be (1) creating a variance-aware filtered evaluation set with some existing systems and then (2) using the new evaluation set to evaluate new systems.*
>
> To alleviate your doubt, we calculate the variance over a random half of the submitted systems and evaluate the generated test set on the remained half, as shown in the following table. For example, the WMT20 Chinese-English (Zh-En)  task has 16 submitted results, we use a random half of the results (*dong-nmt.1207, Tencent_Translation.1249, WMTBiomedBaseline.183, Online-G.1569, DeepMind.381, Huawei_TSC.889, Online-Z.1646, WeChat_AI.1525*) for calculating variance and generating a variance-aware test set, and then evaluate the remained results (*DiDiNLP.401, SJTU-NICT.320, Online-A.1585, zlabs-nlp.1176, Huoshan_Translate.919, OPPO.1422, Online-B.1605, THUNLP.1498*) over the original test set and variance-aware test set.
>
> *Table Ⅴ. Pearson correlations using original and variance-aware test sets on the most competitive WMT20 Chinese to/from English tasks. Using variance-aware test sets generated by a random half of systems (+Half) consistently improves the evaluation results on the remained systems.*
>
> |         |   Zh-En   |   En-Zh    |
> | ------: | :-------: | :--------: |
> |    BLEU |   0.957   |   0.888    |
> | *+Half* | **0.961** | **0.893**  |
> |   COMET |   0.953   |   -0.840   |
> | *+Half* | **0.963** | **-0.821** |
> |  BLEURT |   0.941   |   0.803    |
> | *+Half* | **0.955** | **0.811**  |
> | BERTS-P |   0.938   |   0.902    |
> | *+Half* | **0.947** | **0.948**  |
> | BERTS-R |   0.956   |   0.934    |
> | *+Half* | **0.965** | **0.947**  |
> | BERTS-F |   0.949   |   0.924    |
> | *+Half* | **0.958** | **0.948**  |
>
>
> The results show that the proposed method is still effective in such a scenario. Since the aim of this paper is to provide new test sets for the community to conduct evaluation for new systems, we calculate the variance of all submitted results to make the calculated variance more reliable and the generated test sets more discriminative. We will include the table and related discussion in the revised version to make the proposed method more convincing. Thank you for the questions.

---

> > ### Comment · Area_Chair_2K1E · 2021-07-21
> > **reply on additional results**
> >
> > Thank you for providing additional results to address my last-minute question!

---

### Decision · Program_Chairs · 2021-07-26

**Decision:**

Accept

**Comment:**

The paper presents a new evaluation dataset for machine translation, focusing on sampling test examples that can effectively rank different systems. The evaluation set contains a wide range of language pairs (35 language pairs) and shows a higher correlation with human judgments than original, unfiltered datasets.

All reviewers agreed that the paper points out a new perspective on the MT task, and the same idea of filtering evaluation datasets to improve discrimination ability can be applied for other tasks. The paper also includes in-depth analysis, such as what kind of sentences are getting filtered, and how it connects to human paraphrasing. My initial concern about using same models to compute variance and then to evaluate output was addressed during the rebuttal period.